# What Levers to Promote Teachers’ Wellbeing during the COVID-19 Pandemic and Beyond: Lessons Learned from a 2021 Online Study in Six Countries

**DOI:** 10.3390/ijerph19159151

**Published:** 2022-07-27

**Authors:** Nathalie Billaudeau, Stephanie Alexander, Louise Magnard, Sofia Temam, Marie-Noël Vercambre

**Affiliations:** 1MGEN Foundation for Public Health, 75015 Paris, France; nbillaudeau@mgen.fr (N.B.); salexander@mgen.fr (S.A.); stemam@mgen.fr (S.T.); 2Education and Solidarity Network, 1000 Brussels, Belgium; lmagnard@mgen.fr

**Keywords:** teachers, school, wellbeing, job satisfaction, life satisfaction, work/life balance, mental health, social support, COVID-19, psychosocial factors

## Abstract

To highlight effective levers to promote teachers’ wellbeing worldwide, particularly during difficult times such as the COVID-19 pandemic, we investigated work-related factors associated with teacher wellbeing, across borders and cultures. In six countries/territories, we examined the factors that were most consistently and strongly associated with two indicators of wellbeing at work: (i) job satisfaction; and (ii) work/life balance, and three indicators of general wellbeing: (i) subjective health; (ii) mental health; and (iii) life satisfaction. Between May and July 2021, after 18 months of the pandemic, 8000 teachers answered the first edition of the International Barometer of Education Personnel’s Health and Wellbeing (I-BEST): 3646 teachers from France, 2349 from Québec, 1268 from Belgium, 302 from Morocco, 222 from The Gambia, and 215 from Mexico. For each country/territory and each wellbeing indicator, we used a forward stepwise regression procedure to identify important determinants among a carefully selected set of 31 sociodemographic, private, and professional life factors. Aside from healthcare access, the factors most consistently and strongly associated with teacher wellbeing in France, Québec and Belgium (samples whose size were ≥1000) were related to the psychosocial and the organizational dimensions of work, namely: feeling of safety at school, autonomy at work, and the quality of relationships with superiors and quality of relationships with students. In the smaller samples of teachers from the three remaining countries (Morocco, The Gambia and Mexico), exploratory analyses showed that the feeling of safety and autonomy at work were, there too, consistently associated with wellbeing indicators. During the COVID-19 pandemic, the factors most consistently associated with teachers’ wellbeing across countries were related to security and autonomy at work, supporting the importance to consider these aspects in a continuous, structural way at school. Factors associated with teachers’ wellbeing in very different contexts require further cross-cultural study.

## 1. Introduction

Teachers represent a large working population of more than 90 million individuals all around the world [1]. Teacher wellbeing is a key factor of students’ academic achievement, but more globally, of the wellbeing of tomorrow’s adults [2,3,4,5], and as such, is a major asset for society. During a crisis, whatever its type (security, health, economic etc.), the promotion of teachers’ health is even more important as they act as “shock absorbers” for young people and ultimately for society as a whole [6].

### 1.1. Teaching during the COVID-19 Pandemic

The COVID-19 pandemic has represented a particularly challenging time for education staff: from March 2020, most schools around the world had to close suddenly and staff were required to set up remote learning with no real preparations, and frequently, inadequate resources [7,8]. One year later, more than 800 million learners (half of the world’s student population) were still affected by full or partial school closures. Two years later in March 2022, despite the new omicron variant wave, specific health/safety protocols and/or vaccination coverage allowed most schools to re-open. However, learning losses and effects on students’ health and wellbeing are expected to be felt for many years [9,10]. For teachers, the series of school closures/reopenings with incessantly changing protocols and the fluctuating fear of being infected or infecting others proved to be particularly challenging [6]. Furthermore, the complexity of hybrid teaching weighed heavily on their everyday lives; it involved a mix of in-school and distance teaching (possibly with their own children at home), or in-school teaching with only a selection of students present, while others remained at home [11,12,13,14,15,16,17]. Additionally, some COVID-19 mitigation measures, such as mask wearing or student contact restrictions, although clearly essential to make school a safe place during the pandemic [18], affected teaching methods and the diversity and richness of educational exchanges [19]. All of this has strongly affected the wellbeing of the entire education community and has led to the urgent promotion of one of its essential components: teachers’ wellbeing.

### 1.2. Important Determinants of Teachers’ Wellbeing: What Does the Literature Say?

Several levers have already been put forward to address teachers’ wellbeing [20,21,22,23], in particular helping them build social relationships, providing social support at work, facilitating communication among members of the education community [24,25,26,27,28,29,30], as well as promoting autonomy/self-efficacy, training opportunities, and fostering empowerment [31,32,33,34,35]. In a recent systematic review including 84 quantitative and mixed-methods studies of teachers’ wellbeing correlates from various disciplines [20], the authors noted a high level of heterogeneity, not only in the definition of teacher wellbeing, but also regarding potential determinants investigated; each study considering its particular set of covariates. Among the broad range of variables that were related to teachers’ wellbeing, they distinguished between “general” and “job-related” categories at the “individual” or the “contextual” level, and concluded that although both objective and subjective aspects played a role in teacher wellbeing, the influence was clearer and more pronounced for subjective factors. Among them, social relationships were highlighted as playing a “pivotal role”. The authors underlined the under-representation of studies considering the specific demands of the teaching profession as potential determinants (e.g., relationships with students), as well as of culture-comparative studies. Furthermore, they mentioned that most available studies focused on a specific group of teachers (e.g., secondary school teachers), and/or teachers from one specific country and/or a specific aspect of wellbeing (e.g., negative affect), yielding findings of limited scope and generalizability. These limitations call for further data and investigation of wellbeing as a multidimensional construct, taking into account potential differences across cultures.

### 1.3. Research Gap, Objectives and Conceptual Framework

Teacher wellbeing is at a crossroads of different themes (i.e., educational science, occupational health, public policy etc.), disciplines (i.e., sociology, psychology, ergonomics, epidemiology, philosophy etc.), and cultures and as such, differences in vocabulary, theoretical approaches and methods render it difficult to capture a clear and global view of the field [36]. Although plethoric at first sight, data are scattered, both in time and space, and are produced mostly from cross-sectional or short-term longitudinal studies [20], which are now potentially outdated by the COVID-19 crisis [37]. Furthermore, available studies focused on a specific category of factors and/or one specific indicator of wellbeing. 

In order to gain a more global vision of the levers to promote teachers’ wellbeing after 18 months of the COVID-19 pandemic, we statistically investigated the factors that were most consistently and strongly associated with five indicators of wellbeing in different countries or territories. To complement more conceptual points of view focusing on a single category of determinants in a single territory, we simultaneously considered different categories of potential determinants related to private or professional life, some conjunctural, other structural, some individual, other systemic. 

Our study is informed by the “Conceptual framework of teachers’ occupational well-being” developed by the Organisation for Economic Cooperation and Development (OECD) [38]. Globally, the OECD conceptual framework defines teacher’s occupational wellbeing by drawing on four components, including physical and mental wellbeing, cognitive wellbeing, subjective wellbeing and social wellbeing, all of which the authors propose are shaped by teachers’ working conditions, both at a system level (e.g., working hours, earnings, professional development etc.), and at the school level (e.g., quality of relationships with school staff, work autonomy, learning environment etc.) [38]. All of this, they suggest, has various outcomes for teachers regarding their willingness to stay in the profession and their levels of stress, as well as the quality of learning environments for students and their wellbeing [38]. In line with this work, our epidemiological methodology draws on these wellbeing concepts to inform our definition of wellbeing, our choice of indicators used to measure and analyze teacher wellbeing, and our analysis of factors most consistently associated with them. Consistent with more holistic approaches [20], and because a teacher is first and foremost a human being with basic needs [39], we have also gone beyond the OECD teachers’ occupational wellbeing framework [38] by considering some indicators of general wellbeing and quality of life (wellbeing indicators without explicit reference to work) [39].

Our research question was: “are there work-related factors that would be associated with teacher wellbeing, across borders and cultures, and if so, which ones?”. Our hypothesis was: “Certain work-related factors are consistently associated with teachers’ wellbeing, and these relate to basic needs such as security, autonomy and social relationships”. 

In highlighting the consistently significant determinants of teachers’ wellbeing, we aimed to identify important levers for promoting the wellbeing of educational communities at an international scale, especially relevant during the COVID-19 pandemic.

## 2. Materials and Methods

### 2.1. International Barometer of Education Personnel’s Health and Wellbeing (I-BEST)

The International Barometer of Education Personnel’s Health and Wellbeing (International Barometer of Education Staff ( I-BEST)) is a biennial multi-territory online survey of education staff that was launched in 2021 by the Education and Solidarity Network (ESN, https://www.educationsolidarite.org/en/home/, accessed on 15 June 2022) and the MGEN Foundation for Public Health (https://www.fondationmgen.fr/, accessed on 15 June 2022). This barometer relies on a non-probability sampling procedure mobilizing ESN partners around the world: health mutual benefit societies, education unions, and the UNESCO Chair Global Health and Education (https://unescochair-ghe.org/, accessed on 15 June 2022). It aims to provide timely data on work conditions and the health and wellbeing of education staff, to identify areas for improvement, and to share best practices across borders (https://www.educationsolidarite.org/en/how-are-teachers-feeling-around-the-world-launch-of-the-international-barometer-on-education-personnels-health-and-well-being/, accessed on 15 June 2022).

The first edition of the Barometer targeted in-service teachers whose students were aged 3 to 18 from 6 countries or territories where local ESN partners were interested and able to disseminate the survey at the appointed time. The countries/territories included in this edition were: Québec and Mexico in the Americas, France and the French Community of Belgium/Federation Wallonia-Brussels (henceforth referred to simply as “Belgium”) in Europe, and Morocco and The Gambia in Africa. In practice, the local partners distributed the survey in May, June or July 2021 (depending on the country/territory) and sent a reminder, by directly mailing the country-specific URL link to teachers available in their databases. The dissemination of the survey also benefited from more general publicity via social media (posts, articles etc.) and newsletters whose audience was mainly education staff. The first few questions of the survey made it possible to filter out participants who were not within the target population. The questionnaire, available in French, Spanish, Arabic and English, included approximately 100 questions organized into 8 sections: physical and mental health; professional experience; working conditions and environment; organization and school administration; relationships; conditions, information and support regarding health; experience regarding health; and socio-demographic characteristics.

At no time during the completion of the questionnaire was a name or any identifying information collected from the participants. Because the I-BEST is an anonymous and voluntary survey, the General Data Protection Regulation (GDPR, Europe’s data privacy and security law) did not apply, and informed consent was not required, as confirmed by the MGEN Institutional review board of data protection officers.

In total, around 8000 teachers answered the questionnaire: 1268 teachers in Belgium, 3646 in France, 2349 in Québec, 215 in Mexico, 302 in Morocco, and 222 in The Gambia. The samples of respondents in Belgium, France, and Québec were large enough to be redressed by weighting, referring to the teacher statistics at the territory-level, so that the sample would be representative in gender and teaching level in Québec (primary/secondary school), and in gender, age group, and teaching level in France and Belgium.

### 2.2. Wellbeing Defined 

Wellbeing is a complex, multi-factorial concept linked to satisfaction with life, and which includes physical, material, social, emotional, developmental, and activity dimensions [40]. Diener, Lucas & Oishi [41] provide a short definition of wellbeing as a “general term covering how well individuals are doing in life, including social, health, material, and subjective dimensions of well-being” (p. 3). In particular, subjective wellbeing, defined as the types of subjective evaluations of one’s life, including cognitive evaluations and affective feelings [41], is strongly related to, but not limited to, subjective health [42]. Given its diversity and the lack of consensus regarding a definition [20], it requires a multidisciplinary understanding. 

Particularly relevant to this study, we broadly drew on Viac and Fraser’s [38] four-dimensional framework and operationalization of teacher wellbeing. Viac and Fraser’s framework for teacher wellbeing includes: (1) a cognitive dimension, operationalized by the indicators “capacity to concentrate at work”, and teacher “self-efficacy” (p. 24); (2) a subjective dimension, operationalized by indicators on “satisfaction with current job and the teaching profession”, the “frequency of moods and emotions with regard to job activities”, “purposefulness”, and “satisfaction with life” (p. 25); (3) a physical and mental dimension, operationalized by the “frequency of psychosomatic symptoms during the school year as pertaining to work”, and “number of school days missed due to these symptoms” (pp. 26–27); and (4) a social dimension, operationalized by the “social function in relationships with principals, colleagues, and students”, and “feelings of trust” (p. 28) [20,38].

### 2.3. Wellbeing Indicators

Employment, as an important component of adult life, has a two-way relationship with wellbeing, both influencing and being influenced by it. Wellbeing at work is the component of general wellbeing that is most closely linked with a work context, including job satisfaction and sense of purpose at work [43,44]. In order to evaluate individual wellbeing in its multidimensionality [45,46] among teachers in the present study, we focused on five indicators of wellbeing available in the Barometer chosen for their complementarity: two indicators of wellbeing at work (job satisfaction and work/life balance), and three indicators of general wellbeing (subjective health, mental health, life satisfaction). Each indicator is described below. Pearson’s correlation coefficients were computed between indexes of wellbeing to confirm that although linked, they were not redundant (Appendix A).

(a)Wellbeing at work

First, we computed an indicator of job satisfaction (score range: 0–9) by summing three responses on a Likert scale that have frequently been used in the OECD Teaching and Learning International Surveys (https://www.oecd.org/education/talis/, accessed 15 June 2022, see also: [47]): “To what extent do you agree or disagree with the following statements?”: (1) “The advantages of being a teacher more than compensate the disadvantages.”, (2) “If I had to do it again, I would choose teaching again.” And (3) “Overall, I am satisfied with my job”. Responses ranged from 0 = “strongly disagree” to 3 = “strongly agree”. We defined “Job satisfaction” as having a score of ≥6.

Second, an indicator of work/life balance [48] was based on the question “Do you feel that your work drains so much energy and/or takes so much time that it has a negative impact on your private life?”. Response items included: “never”, “rarely”, “sometimes”, “often”, “very often”, and “always”. Participants who responded “always” were considered to have a poor “work/life balance”.

(b)General wellbeing

We created the indicator of subjective health with the question: “In general, would you say that your health is…”. The response items included: “excellent”, “very good”, “good”, “fair”, “rather poor”, and “poor”. Participants who responded with “rather poor” or “poor” were considered to have poor “subjective health”.

Then, we focused on the mental health component of general wellbeing and referred to the question “How often do you have negative feelings like blues, anxiety, depression, hopelessness?”. Response items included: “never”, “rarely”, “sometimes”, “often”, “very often”, and “always”. Participants who responded “often”, “very often”, or “always” were considered to have poor “mental health”.

Third, to complete the analysis, and as an alternative outcome of general wellbeing, we considered an indicator of life satisfaction using a visual analog scale (picture of a ladder) where respondents were asked to situate their level of life satisfaction from “best possible life” (scored 1) to “worst possible life” (scored 8). “Life satisfaction” was defined as having a score = 1, 2 or 3. 

### 2.4. Potential Determinants of Wellbeing

As potential determinants of wellbeing, we considered a list of 31 factors previously linked to a dimension of wellbeing in the general population, or more specifically among teachers [20,21,37,49,50,51,52,53,54], and that were available in the Barometer. In addition to sociodemographic characteristics (age and gender), we considered factors related to private life (household partner; household child(ren); access to healthcare; access to training/prevention resources) and to professional life. Regarding professional life, we considered general factors (age of students; seniority; employment status; union membership; remote work; school sector), but also factors related to the physical environment (school size, school urbanicity, school facilities; material conditions; indoor air quality; indoor noise level), the psychosocial environment (feeling of safety at school; victim of violence; witness of violence; quality of relationships with students; quality of relationships with colleagues; quality of relationships with superiors), and the organization/career perspectives (autonomy at work; teamwork; communication; societal appreciation; salary; continuing education; career progression). The survey questions used to define the 31 factors considered in the present analysis, as well as how each factor was introduced in the statistical models (including the reference group), are outlined in the Appendix A.

### 2.5. Statistical Analysis

First, we described the teachers’ main characteristics and the five wellbeing indicators across the six countries/territories participating in the 2021 Barometer (France, Québec, Belgium, Morocco, The Gambia, Mexico). Secondly, for the analytic approach of wellbeing, in order to have sufficient statistical power for the 31 potential wellbeing determinants under study, we focused on the three larger teacher samples, those whose sample size was ≥1000. This included France, Québec and Belgium. Alternately for each dichotomous wellbeing indicator (job satisfaction, poor work/life balance, poor subjective health, poor mental health, life satisfaction), and for each country sample, we ran forward stepwise regression analyses to automatically select a parsimonious set of important wellbeing factors according to the national context [55]. It is noteworthy that the results of any stepwise regression procedure should not be overstated because of methodological issues [56], but as our approach consists of a preliminary scoping of teachers’ wellbeing determinants, this intuitive procedure remains informative.

By way of synthesis, for each potential determinant and each wellbeing indicator, we calculated the consistency (defined as the number of association(s) that were found to be significant among the three models of the wellbeing indicator performed alternatively in France, Québec and Belgium), and the mean effect (mean of the corresponding significant beta(s)). Then, for each potential determinant, we computed a synthetic impact score on wellbeing as the average of the mean effect weighted by the consistency across the five wellbeing indicators studied. In this synthesis step, the five dimensions of wellbeing were carefully coded with the same valence, namely positive, opposing when needed the effects estimated in the dichotomous “negative valence model”. “Wellbeing” was then appraised through the composite indicator giving equal weight to the five dimensions of job satisfaction, work/life balance, subjective health, mental health, and life satisfaction. Analyses were run using STATA version 17 (StataCorp. 2019. Stata Statistical Software: Release 16. StataCorp LLC, College Station, TX, USA).

## 3. Results

### 3.1. Teachers’ Characteristics and Wellbeing across 6 Countries/Territories

Samples of teachers from the six countries/territories participating in the 2021 Barometer include a variety of profiles (Table 1).

Over 70% of the teachers who responded were women, except in The Gambia where men represented 68% of the sample. The different teaching levels were well represented in the countries/territories surveyed, except in Mexico where primary school teachers were largely predominant, whereas they were very low in number in The Gambia. The samples included teachers with varying seniority, the intermediate class of “5–30 years of experience” was the largest and the most represented, and the average teacher age was approximately forty years in all countries/territories except in The Gambia (respondents there were much younger). Finally, in conjunction with the local stage of the COVID-19 pandemic at the time of the survey, 100% of Mexican teachers who responded were teaching remotely on at least a partial basis (80% on a total basis), and 51% in The Gambia, whereas more than 60% were teaching entirely on-site in the other countries.

Regarding indicators of wellbeing at work (Figure 1) or general wellbeing (Figure 2), they appeared highly mixed across the six countries/territories studied. However, the differences in size and composition of the six samples (Table 1) preclude any direct comparison.

Overall, subjective health appears to be preserved with fewer than two out of ten teachers considering it to be rather poor or poor (Figure 2). However, the mental dimension of health was particularly affected in France, Belgium and, to a lesser extent, in Québec, with almost one out of two teachers reporting often, very often or always experiencing negative feelings.

### 3.2. Covariates of Wellbeing at Work among Teachers in France, Québec and Belgium

The logistic regression models selected by the stepwise procedure in the French, Québec and Belgian samples respectively are outlined in Appendix A for job satisfaction and in Appendix A for work/life balance. Overall, the factors whose impact measures are always highest regarding wellbeing at work, whether in terms of consistency or of mean effect, are the psychosocial factors at work, namely: the feeling of safety at school and the quality of relationships with students, with colleagues (for work/life balance) and with superiors. Additionally, two factors involving the organizational/career perspectives were consistently and closely associated with a higher level of wellbeing at work: autonomy at work and salary.

### 3.3. Covariates of General Wellbeing among Teachers in France, Québec and Belgium

Concerning the models of the three indicators of general wellbeing (subjective health: Appendix A, mental health: Appendix A, and life satisfaction: Appendix A), sociodemographic and private life factors were found to be more frequently associated with these indicators than with the indicators of wellbeing at work. In particular, “access to healthcare” and “access to training/prevention resources” were both consistently associated with higher levels of general wellbeing.

Regarding the professional life factors, the same factors previously highlighted as important determinants of wellbeing at work emerged as similarly important for general wellbeing: the feeling of safety at work, the quality of social relationships at school and autonomy at work. Additionally, violence at work was consistently associated with lower levels of general wellbeing.

### 3.4. Synthesis in France, Québec and Belgium

When considering consistencies, mean effects and synthetic impact score (IS) side by side (Figure 3), the factors with the highest IS were: the feeling of safety at school (IS = 2.7), the quality of relationships with students (IS = 1.7), the quality of relationships with superiors (IS = 1.9), and autonomy at work (IS = 1.9). Additionally, but to a lesser extent (0.5 < |IS| < 1.5), healthcare access (IS = 1.1), the quality of relationships with colleagues (IS = 0.7), violence at school (IS = −0.6), salary (IS = 1.0), and the career progression (0.7) were all consistently associated with teachers’ wellbeing.

### 3.5. Exploratory Analysis among Teachers in Morocco, The Gambia and Mexico

To evaluate to what extent the results observed among teachers in France, Québec and Belgium could be generalized to other contexts, we carried out exploratory analyses in countries/territories with smaller teacher samples (Morocco, The Gambia, Mexico). Overall, these results confirmed the importance of the feeling of safety at school and of autonomy. However, in these rather different settings, the quality of relationships with students and superiors was no longer consistently associated with wellbeing indicators.

## 4. Discussion

After 18 months of the COVID-19 pandemic, teachers’ wellbeing as evaluated in the six countries/territories participating in the International Barometer of Education Personnel’s Health and Wellbeing (I-BEST) varied widely. In the analysis of its major covariates performed in the samples whose size was ≥1000 individuals (i.e., France, Québec and Belgium), apart from a factor not directly related to work—namely, healthcare access in the country—the factors most consistently and closely associated with teacher wellbeing involved the psychosocial and organizational dimensions of work: the feeling of safety at school, the quality of relationships with students, the quality of relationships with superiors and autonomy at work. In the smaller samples (i.e., Morocco, The Gambia and Mexico), exploratory analyses showed that the feeling of safety and autonomy at work were, there too, consistently associated with wellbeing indicators. These results partly support our hypothesis: the professional factors most consistently associated with teacher wellbeing across countries are related to basic needs, in particular security and autonomy. That social relationships at work were less consistently associated with teachers’ wellbeing in very different national backgrounds requires further cross-cultural study.

### 4.1. What Lessons Learned about Teachers’ Wellbeing across 6 Countries/Territories in 2021?

The six countries/territories involved in the first edition of the Barometer have very different socio-economic, political and cultural backgrounds. Additionally, at the time the survey was delivered, the COVID-19 situation differed strongly depending on the location, as is illustrated by the disparate rates of teachers still working remotely when responding to the survey (from 23% in Belgium to 100% in Mexico). In addition, the differences in sample size (more than 1000 individuals for 3 countries/territories, less than a few hundred for the three others) and sample composition (e.g., gender, age, grade level taught), precludes direct comparisons or hasty conclusions. For France, Québec, and Belgium, as each sample included more than 1000 individuals with varied profiles, we were able to adjust the data by weighting, so that descriptive statistics for these three countries/territories would be more generalizable to the national level. All these methodological, structural, and situational differences must be taken into account when interpreting raw statistics. Overall, at the end of the 2020/2021 academic year, after a long pandemic period alternating between in-person and remote teaching, and with changes between stricter and more relaxed health protocols [57], teacher wellbeing appeared highly mixed across countries/territories and rather concerning in its mental dimension in France, Belgium, and Québec, as already suggested by various studies in Europe or North America [11,15,17].

### 4.2. What Lessons Learned about the Levers Most Consistently Associated with Teachers’ Wellbeing across Countries/Territories?

As compared to the descriptive analysis, the analytic approach at the core of this work aiming to identify important levers of teachers’ wellbeing is less vulnerable to response bias [58]. Two points should nonetheless be kept in mind before outlining practical implications. First, the study is cross-sectional, precluding the interpretation of statistical associations as causal. Notably, some associations may in fact be bi-directional, such as the association between the quality of relationships with students and mental health [59]. Secondly, it is not self-evident that results observed in three French-speaking countries/territories with rather similar cultural and socioeconomic backgrounds could be generalized to other contexts. To evaluate the robustness of our results, we carried out exploratory analyses in the smaller teacher samples (i.e., Morocco, The Gambia, Mexico), and the analyses in these countries with very different backgrounds to some extent confirmed our main analysis, supporting our finding that the feeling of safety at school and autonomy at work are important and possible universal determinants of teachers’ wellbeing. This is also in line with previous studies among teachers illustrating the importance of autonomy [21,31,60,61,62], and of a safe/peaceful environment [63,64,65]. However, some of these studies were conducted before the COVID-19 pandemic and all were conducted in a specific national context.

In addition to bringing updated data from several different countries, we were able to consider many potential wellbeing determinants simultaneously, in particular occupational factors, which allowed us to evaluate the effect of each factor independently of the others. Overall, our study supports the evidence establishing safety and autonomy at work as key factors of the wellbeing of employees [50,66]. Furthermore, our results contribute to the large body of literature that more generally posits the importance of autonomy as one of the primary psychological needs for human beings, and that this is fundamental to overall wellbeing [67,68,69].

The fact that relationships with students and superiors were not consistently associated with teacher wellbeing in Morocco, The Gambia, and Mexico suggests that the psychosocial context of the work environment may be less of a priority in these countries than other aspects such as the physical environment. Indeed, in the exploratory models selected by the stepwise procedure in these three countries, material conditions and school facilities were repeatedly found to be significantly associated with one or the other wellbeing outcome, even though the small sample size renders it more difficult to detect significant associations. Although exploratory, these observations point to the importance of thoroughly considering the country context using updated reliable local data before implementing any health promotion actions or programs.

Interestingly, in the specific context of the survey, after several waves of the COVID-19 pandemic, remote working did not appear to be an important determinant of teacher wellbeing. Indeed, across the five wellbeing indicators studied, only one (life satisfaction) was found to be significantly (and negatively) associated with working remotely, and only in one country (Belgium). The fact that this was found in only one country was rather surprising, as full-time remote work has been linked with increased stress, especially among women [70], including in the education sector [14]. In fact, the proportion of teachers working from home all of the time was so small in these three samples (2% in France, 5% in Québec, 1% in Belgium) that they had to be considered together with those working only partially from home. Yet, these two situations are rather heterogeneous, especially in terms of health status and mental health impact, so that the present study does not allow us to draw definitive conclusions on this point.

### 4.3. Practical Implications

Noteworthy is the “broadness” of the determinants of teachers’ wellbeing highlighted here: “safety”, “autonomy”, and “social relationships” are general factors with various scopes and many underlying levers, but the most effective lever in one country/territory will not necessarily be relevant in another [62]. Rather than privileging a top-down strategy, it will be important that each education system, and each school within the system, discusses which levers best fit their specific situation. The present study may be used as a starting point for discussion. As a general rule, research and interventions to address health and wellbeing need to focus not only on individual level factors but also on organizational and societal-level factors given their close interrelation [49].

For example, to improve “safety at school”, it would first be necessary to clarify which form of “safety” is involved: safety defined as the absence of physical or psychological violence/terrorism (“security”), or as the absence of microbial exposure (“microbial safety”), or of chemical or physical risks (“occupational safety”)? The hypothesis that microbial safety was particularly important for this first edition of the Barometer is supported by the heightened coronavirus threat at the time of the survey, but this remains to be confirmed. If this were the case, the strategy could include reinforced sanitary protocols and mitigation measures, such as wearing a mask, social distancing, frequent hand washing, no mixing of student classes, and so forth [71]. For instance, the national level could provide decisive impetus for the strategy, leaving room for adaptation at the local level, which could also continually adapt any measures to the changing circumstances.

### 4.4. Limitations and Perspectives

Overall, study limitations should be considered when interpreting our results. The cross-sectional design, the non-probabilistic sampling and the difficulty to generalize the results have been discussed above. Another limitation is that data are self-reported. Nonetheless, the data remain convenient and informative in the present analysis of subjective wellbeing, as the effect of potential stressors importantly depends on how they are appraised by the individual [72]. Furthermore, the sample sizes and the way the outcome is defined, affect the magnitude and the significance of the observed associations. The use of stepwise regression, although an intuitive and interesting approach when dealing with several samples and numerous potential determinants, requires caution in interpretation [56]. In addition, we were not able to consider all potential wellbeing determinants and possible interaction effects (those implying personality traits or emotion regulation strategies) even though they may play an important role in the process [26,37,73,74]. Moreover, information on certain categories of teachers or education staff is lacking (e.g., university teachers or school support staff). Finally, as mentioned earlier, wellbeing is a multifaceted concept and it cannot be entirely apprehended in its complexity by five, albeit complementary, indicators [75]. In fact, our approach, although quantitatively based, should be considered qualitative and exploratory in its conclusions. The next edition of the Barometer, scheduled for 2023 and to be completed in additional countries/territories, will target all school staff and not only school teachers. It will allow us to delve further into the subject, confirm or refute the associations presently highlighted, and evaluate trends. To decisively strengthen the body of evidence, longitudinal and intervention studies would also be crucial [20,37,76]. 

## 5. Conclusions

Drawing on an online survey of 8000+ teachers completed in 2021, we evaluated teachers’ wellbeing in six countries/territories after eighteen months of COVID-19 pandemic and investigated its main covariates. In highlighting the consistently significant determinants of teachers’ wellbeing across countries/territories, we aimed to identify important levers for promoting the wellbeing of educational communities at an international scale. 

Beyond the heterogeneity of teacher wellbeing around the world, in the national samples of teachers whose size exceeded 1000 (in France, Québec and Belgium), we observed that factors most consistently and strongly associated with teacher wellbeing were: 1/ feeling of safety at school, 2/ autonomy at work, and 3/ quality of relationships with superiors and with students. In the exploratory analyses of the smaller samples of teachers (Morocco, The Gambia and Mexico), feeling of safety and autonomy at work were also consistently associated with wellbeing indicators, whereas quality of relationships with superiors and students were not. Using updated data from various territories, our epidemiologic study considered a great number of potential determinants of wellbeing simultaneously, complementing more conceptual points of view focusing on a single category of determinants. Limitations of our study included the non-probabilistic sampling, its cross-sectional design, the self-reported nature of the data, the possible impact of various sample sizes across countries, and pragmatic choices of variables based on availability. Our study should be considered exploratory in nature, and as serving as a basis for discussion by highlighting important levers for promoting teacher wellbeing at school. 

In the particular context of the COVID-19 pandemic, our results contribute to the body of knowledge on determinants of teachers’ wellbeing, highlighting as promising levers, both safety and autonomy at work. Our results suggest that these two factors should remain a priority at school in a continuous, structural way, to prevent consequences linked to a variety of crises. Other factors related to relational climate or the physical environment can make a difference in the long term, depending on the local background and circumstances. 

The next edition of the Barometer, scheduled for 2023, will include additional countries/territories and target both teachers and non-teaching school staff, allowing us to replicate and hopefully expand the present analysis. To decisively complete this multi-territory, cross-sectional approach and inform healthy school research and policy, longitudinal studies that concomitantly investigate the personal and contextual, work and non-work determinants of teachers’ wellbeing should be carried out, as well as intervention studies in schools targeting the wellbeing determinants highlighted.

## Figures and Tables

**Figure 1 ijerph-19-09151-f001:**
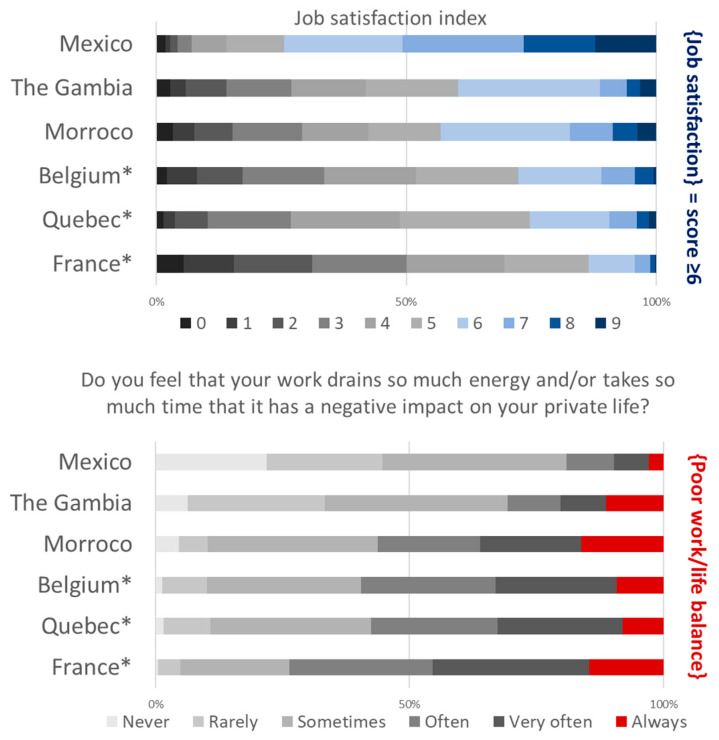
Two indicators of teachers’ wellbeing at work: job satisfaction, work/life balance, 2021 International Barometer of Education Personnel’s Health and Wellbeing (I-BEST). (* Weighted statistics taking into account gender and teaching level + age group in France and Belgium).

**Figure 2 ijerph-19-09151-f002:**
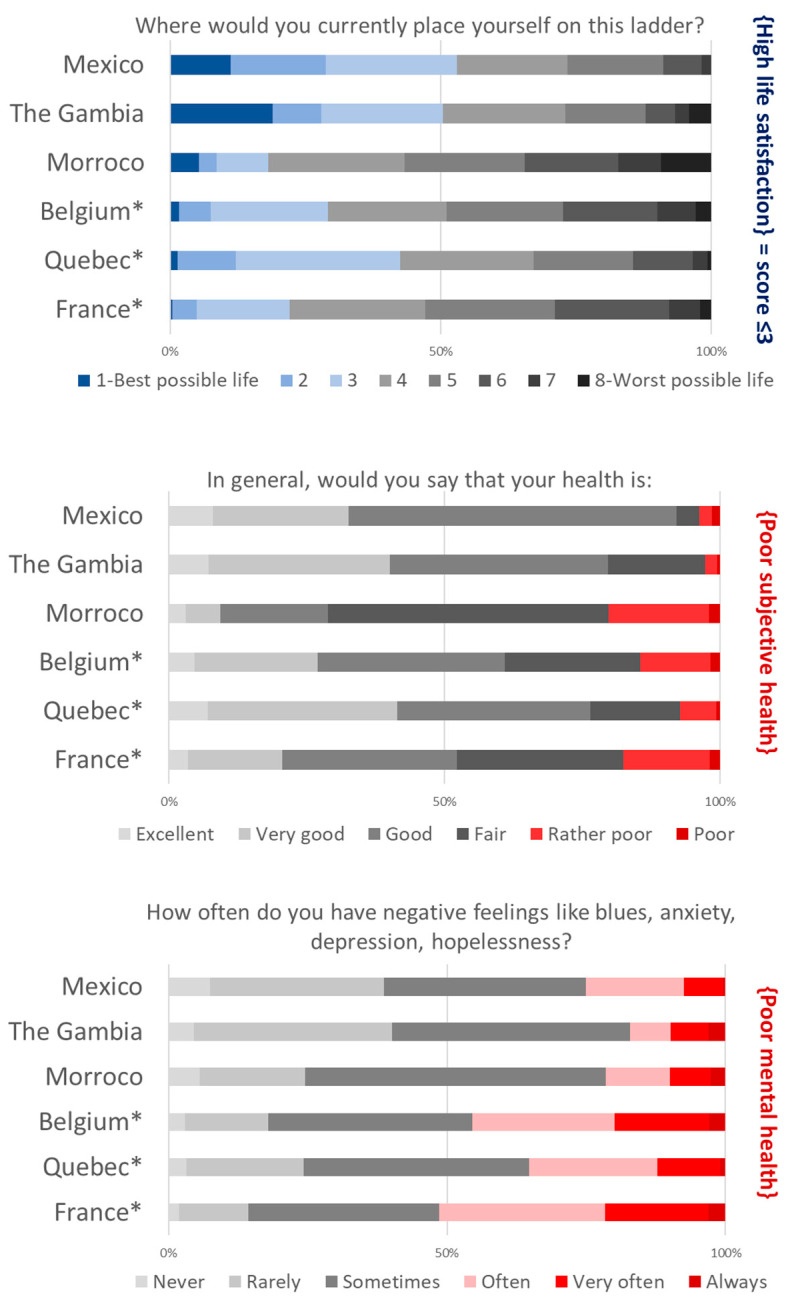
Three indicators of teachers’ general wellbeing: life satisfaction, subjective health, mental health, 2021 International Barometer of Education Personnel’s Health and Wellbeing (I-BEST). (* Weighted statistics taking into account gender and teaching level + age group in France and Belgium).

**Figure 3 ijerph-19-09151-f003:**
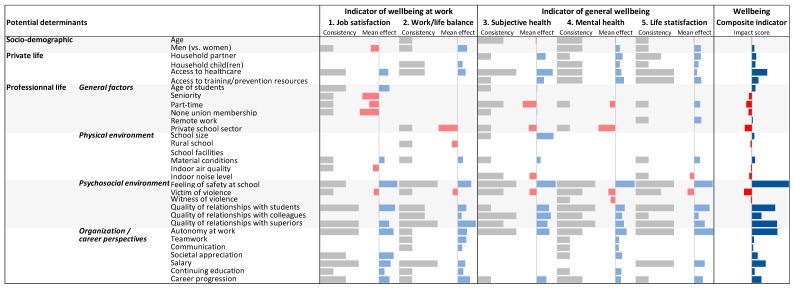
Consistency, mean effect and synthetic impact score for 31 potential determinants of teachers’ wellbeing at work in France, Québec, Belgium, 2021 International Barometer of Education Personnel’s Health and Wellbeing (I-BEST). (Consistency = number of “cofactor × wellbeing-indicator” association(s) that was(were) found to be significant among the 3 wellbeing indicator models performed in France, Québec, Belgium, using a stepwise forward stepwise regression procedure (0 = “null effect observed systematically” to “3 = significant association observed systematically”). Mean effect = mean of the corresponding (significant) beta(s) (red denotes a negative significant effect, blue a positive significant effect, the size of the bar giving the amplitude of the effect). Impact score = average of the mean effect weighted by the consistency across the five wellbeing indicators studied).

**Table 1 ijerph-19-09151-t001:** Characteristics * of teachers participating in the 2021 International Barometer of Education Personnel’s Health and Wellbeing (I-BEST).

	France *	Québec #	Belgium *	Morocco	The Gambia	Mexico
N	3646	2347	1268	302	222	214
Gender (%)						
*Men*	19	15	18	30	68	23
*Women*	81	85	82	70	32	77
Age (median)	44	44	43	43	31	38
Age of students (%)						
*3–5*	18	9	18	1	1	8
*6–11*	28	47	25	39	6	77
*11–15*	29	28	26	28	37	11
*16–18 years old*	25	16	31	31	56	4
Seniority (%)						
*<5*	5	9	6	14	33	20
*5–30*	79	83	79	73	67	72
*>30 years*	16	8	14	13	0	7
Employment status (%)						
*Full-time*	91	89	87	100	94	67
*Part-time*	9	11	13	0	6	33
School sector (%)						
*Public*	94	99	98	98	89	72
*Private*	6	1	2	2	11	28
Union membership (%)						
*Yes*	43	96	83	54	73	67
*No*	51	3	12	36	19	28
*Do not want to answer*	6	1	5	10	7	5
Remote teaching at survey time (%)						
*Yes, totally*	2	5	1	3	13	80
*Yes, partially*	28	31	22	34	38	20
*No*	70	64	77	63	49	0

* Weighted statistics taking into account gender and teaching level (primary/secondary school); # weighted statistics taking into account gender and teaching level (primary/secondary school) and age group.

## Data Availability

The data underlying this article are not publicly available, but the datasets generated and/or analyzed during the current study are available from the corresponding author on reasonable request. This article is made freely available for personal use in accordance with MDPI website terms. You may use, download and print the article for any lawful, non-commercial purpose (including text and data mining) provided that all copyright notices and trademarks are retained.

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
