# Peer review of "What Levers to Promote Teachers’ Wellbeing during the COVID-19 Pandemic and Beyond: Lessons Learned from a 2021 Online Study in Six Countries"

_ijerph, 2022, doi:10.3390/ijerph19159151_

Round 1

Reviewer 1 Report

attached

Reviewer 2 Report

Title: Promoting teachers’ well-being, during the covid-19 pandemic and beyond: lessons learned from an international 2021 online study

Due Date: 8, July 2022

COMMENTS TO THE AUTHORS

________________________________________________________________

Dear Author/s,

Thank you for submitting your manuscript to International Journal of Environmental Research and Public Health.  It is an interesting read on a relevant topic, but there are a number of issues that require your attention before this paper can be published.  Some are minor and can easily be addressed, others require some further reflection and effort, as indicated below:

Your paper needs some improvement.  More specific areas of focus are as follows:

1.     Title and Introduction

1.1  Title

The title of the paper is rather long.  Consider a shorter version that captures the aims of your paper more effectively.  It should not state ‘promoting’ since you are exploring wellbeing and not putting forward suggestions or strategies to promote it.

1.2 Aims, Research Question/s and Hypotheses

The aims of the research are not explicitly pointed out other than in the abstract.  It is suggested that the research aims are more explicit and consistent throughout the document. Moreover, a research question is not clearly articulated.  

1.3 Introduction and Hypotheses Development

There is very little evidence of engagement with the relevant literature. Moreover, the research gap is not clearly articulated in this paper.  It feels that this paper was conceived after that the data was collected.   Kindly articulate the research gap and clearly outline the contribution made through this study to academia.  Defining key terms and use of adequate supporting evidence is highly recommended while avoiding sweeping statements. Some claims are made throughout the document that appear to emerge from assumptions made by the researchers. Such claims could be supported by literature to make the paper more robust. 

Overall the literature review could benefit from better structuring and evidence of proper engagement with the relevant literature through the use of more supporting evidence. 

1.4 Motivation and Contributions

This area is largely missing.  The identification and articulation of a research gap and the addition of the theoretical contribution of the proposed study would make the introduction more robust.

2.     Theoretical Foundations, Model and Hypotheses

The article is missing the conceptual model used to drive the research. This is an important inclusion and it is suggested that the authors give this due consideration.  The conceptual framework should clearly outline the relationship if the constructs explored in the study

3.     Methods

The sampling and selection criteria are not clearly outlined.  Please add more details about these two elements outlining what the criteria for inclusion were in particular.  It is important to refer to the following issues which appear to be missing:

1.     Ethical considerations and ethics clearance obtained,

2.     Issues of anonymity/pseudonymization and how the author/s went about it

Moreover, the authors state that they monitored the quality of responses.  How was this done?  Were responses altered?

4.     Discussion

The discussion section is weak.  Given that there is very little literature to refer to in the introduction/literature review, the emerging results could not be adequately compared to previous research.   Better use of references to substantiate the findings would make the article more robust.

5.     Limitations

This section is missing entirely.  What are the limitations of this study?

6.     Conclusion

What are the implications of this study? How does it add to the current literature in the field? 

General comments.

This paper starts off with a promising abstract. However, the reader is let down due to the lack of structure and academic rigor of the paper. There is lots of potential for improvement.

Firstly, the authors need to engage with the relevant literature and structure the introduction accordingly leading to the identification of a research gap and an appropriate research question.  Proper hypothesis development should be articulated while the theoretical contribution of this work should be identified.

The authors should use less colloquial language.  The procedure of the ‘how’ the data was collected should be kept separate under the Methods section and kept separate from the Introduction.

I hope that you will find the review useful to further develop your work.

Best wishes for further developing your paper.

Reviewer 3 Report

This research debates teachers’ well-being during the COVID-19 pandemic focusing on job satisfaction, work/life balance, and general wellbeing.

The abstract has appropriate length and it includes the purpose of the research, the methods used to achieve the objective of the research, the most important results, and how this research advances knowledge.

The introduction provides adequate framing for the paper, and a sufficient overview of the background to the research questions. The research problem is clearly articulated, with an appropriate rationale and justification of its importance. The research design is clearly described, with adequate justification for the choice of methods and a clear account of how evidence has been analysed. It also demonstrates that acceptable norms of good research practice have been upheld in the conduct of research. The presentation of results is simple and straightforward in style, and clearly laid out and in a logical sequence. This discussion recalls the most relevant findings and is supported by suitable references. The conclusions could be improved by summarising the salient contributions of the paper, and providing some direction for future work, although it demonstrates a firm grasp of the key issues.

My main recommendation was to improve the conclusion. It characterizes the work that was done and highlights why it is important. However, the authors fail to mention: 1. Recalling the reader with a summary of their main findings; 2. Referring to the limitations of their work; 3. Referring to the implications of these results for current knowledge and practice; 4. Be specific enough in identifying how this work can be further developed/improved by other researchers (future work).

Reviewer 4 Report

Dear Authors,

It was a pleasure to read your article. The topic is very actual and extremely important.

1)     My main doubt concerns the lack of a definition of well-being. There is no definition in the introduction. Also, because we do not know what approach has been taken, it is not known how to interpret the results. In psychology, well-being is defined and described in two ways: as a hedonistic, subjective experience of pleasure  or as a eudaimonistic feeling accompanying the realization of human potential. The eudaimonistic approach does not diagnose well-being at a given moment, as a certain effect or some reaction to reality. Instead, it is treated as a stable feature of a person.

This is very important because if we assume that well-being is relatively constant, we can check how people with different well-being levels cope in difficult situations. However, we cannot conclude that the well-being of, for example, teachers should be increased (because it is constant).

A method of researching well-being presented in the text, e.g:

„Third, to complete the analysis, and as an alternative outcome of general wellbeing, we considered an indicator of life satisfaction using a visual analog scale (picture of a  ladder) where respondents were asked to situate their level of life satisfaction from “best possible life” (scored 1) to “worst possible life” (scored 8). “Life satisfaction” was defined as having a score = 1, 2 or 3.”

suggests this approach.

If so, the conclusions should be formulated differently.

Perhaps that's why one of the results is:

„Interestingly, in the specific context of the survey, after several waves of the covid-19 pandemic, remote working did not appear as an important determinant of teacher well- being”.

This result may indicate that well-being is constant and not changeable under the circumstances.

I recommend this article:

Huta, Veronika. (2020). How distinct are eudaimonia and hedonia? It depends on how they are measured. Journal of Well-Being Assessment. 4. 10.1007/s41543-021-00046-4.

To conclude: Please define well-being and interpret the results consistent with the definition.

2)     I am wondering, whether the text relates to the entirety of the research (as the title suggests) or the section on France, Québec, and Belgium (as suggested by the content of the article)?  Please rethink the title

3)     „COVID-19” not „covid-19”. Please use capital letters. 

4)      Please check whether the texts are adapted to the MDPI editing standard.

I hope you will revise the text because the data collected is worth publishing.

Round 2

Reviewer 2 Report

Dear Authors,

I am grateful for the detailed response. I noticed that you took on board most of the feedback.  Important omission that has not been addressed are: conceptual framework, research question and hypotheses that should drive the research.  These should be made more explicit.

Author Response

Dear reviewer, the time and effort you put into your review is very much appreciated. We thank you for your positive view of our previous revision and your further comment on our article. Please, find attached the details on how we took it into acount.

We believe that incorporating these elements makes our manuscript more structured and clearer. Thank you very much for your contribution.

Reviewer 4 Report

Dear Authors,

thank you for the changes made, I accept the text.

Author Response

Dear reviewer, the time and effort you put into your review is very much appreciated. We thank you for your contribution to make our article stronger.

Sincerly.